# The Evaluation of Roasted Lentils (*L. culinaris* L.) Quick Meals as An Alternative to Meat Dishes

**DOI:** 10.3390/foods13010099

**Published:** 2023-12-27

**Authors:** Kristine Ozolina, Ilze Beitane, Vitalijs Radenkovs, Evita Straumite, Anda Valdovska, Sandra Muizniece-Brasava

**Affiliations:** 1Food Institute, Faculty of Agriculture and Food Technology, Latvia University of Life Sciences and Technologies, LV-3004 Jelgava, Latvia; kristineozolina7@gmail.com (K.O.); evita.straumite@lbtu.lv (E.S.); sandra.muizniece@lbtu.lv (S.M.-B.); 2Research Laboratory of Biotechnology, Latvia University of Life Sciences and Technologies, LV-3004 Jelgava, Latvia; vitalijs.radenkovs@lbtu.lv (V.R.); anda.valdovska@lbtu.lv (A.V.); 3Processing and Biochemistry Department, Institute of Horticulture, LV-3701 Dobele, Latvia

**Keywords:** lentils, quick meal, nutritional value, amino acids, vitamins, minerals, sensory properties, texture

## Abstract

Despite the health-promoting benefits, the consumption of lentils in East Europe is low, attracting researchers’ interest in solving the problem. The aim of this study was to develop an alternative to animal proteins for nutrient-dense plant-based quick meals using roasted lentils as the primary raw material, performing sensory analysis, and evaluating the content of amino acids, minerals, and vitamins. The consumption of legumes in Latvia is also low, even though most respondents associate the use of legumes with a healthy choice. Roasted lentil quick meals can deliver 15.6% and 26.2% of the reference intake for protein. Furthermore, one-third of the amino acids (AAs) are essential AAs. AA values in prepared quick meals make them promising alternatives to meat products. One portion of ready-roasted lentils with Bolognese sauce provided above 15% of the daily reference intake of thiamin and vitamin B9. One portion of a ready-quick meal of tomato soup with roasted lentils and roasted lentils with Bolognese sauce provided 20.3% and 25.6% of iron, according to daily reference intake. Further studies on the bioavailability of quick meals must be conducted to claim they can replace meat nutritionally.

## 1. Introduction

The demand for plant-based foods is increasing due to the negative impact of animal-based products on human health and the environment [1]. However, the most serious challenge is to ensure that plant-based protein can meet the human protein requirement as well as animal-based protein. Consequently, there is an increasing emphasis on pulses such as beans, peas, and lentils, which are rich sources of protein—16.7–43.1% and, in addition, contain the valuable essential amino acid lysine [2], because the demand for protein is rising. The consumption of legumes in Europe is low compared to other regions [3]. There are differences between West and East Europe [3], which means that the intake of legumes should be promoted, especially in East Europe, as their consumption is associated with positive health effects [4,5]. Nutrients in legumes can reduce the risk of diseases such as diabetes, obesity, cardiovascular diseases, cancer, etc. and have a beneficial effect on the treatment and prevention of these diseases [6]. In addition, legume proteins are used to produce meat alternatives, but their acceptance by consumers in Western countries is relatively low [7]. However, it is higher than that of cultured meat and insects [8].

Lentils are an annual *Fabaceae* family plant used for food and animal feed, and they play a vital role in sustainable agriculture [2] due to their low environmental footprint [9]. The largest lentil producers in the world are in the Americas and Asia regions, while in Europe, which provides only 1.82% of the total lentils production, the primary producers are France and Spain, while in terms of lentils consumption, Spain is in first place, followed by France and Italy [10,11]. Lentils are rich in protein (24.3–30.2% of dry weight, depending on variety) and low in fat (1.01.3% of dry weight). They also contain various minerals like potassium, magnesium, phosphorus, calcium, iron, zinc, and copper [12], and have a high content of flavonoids [13]. One of the downsides to legumes, including lentils, is their antinutrient content [14], for instance, phytic acid. Phytic acid is a non-protein compound that negatively affects the absorption of protein, fat, and minerals such as iron, zinc, calcium, magnesium, and phosphorus [15]. There is a significant difference in phytic acid content between red and brown lentils—1191 mg 100 g^−1^ of DW and 612 mg 100 g^−1^ of DW, respectively [2]. The phytic acid content in lentils, and accordingly, their negative effect on nutrient absorption, can be reduced by soaking, roasting, and cooking [16]. While the tannins in lentils, whose content is 0.40–1.01% depending on the variety, reduce the protein quality and change the color of the seeds, which can be prevented with heat treatment [12]. Despite the anti-nutritional effect, lentils have a good potential for new product development. The issue is consumer acceptance, as studies indicate that consumption of legumes is low; for instance, only one-third of the Swedish population consumes minimally processed legumes more than once a week [8].

From the consumer side, there is a rapid demand for ready-to-eat, minimally processed [17], quick, and easy-to-make [18] meals as people are increasingly inclined to spend less time in the kitchen. In addition, ready-to-eat products and quick meals do not have to be low in nutritional value; they can be healthy and nutrient-dense, and they are also well-accepted by consumers [19]. Despite the nutritional value and health-promoting benefits of legumes, their consumption in Europe, especially in the East, is relatively low. However, the production of ready-to-eat products and quick meals makes it attainable to increase legume consumption. The aim of this study was to develop an alternative to animal proteins for nutrient-dense plant-based quick meals using roasted lentils as a basic raw material, performing sensory analysis, and evaluating the content of amino acids, minerals, and vitamins.

## 2. Materials and Methods

### 2.1. Study Design

The selection of quick meal products for further analysis of vitamins, minerals, and amino acids has been based on preliminary trials conducted through surveys and sensory analysis (Figure 1).

### 2.2. Survey on Consumption Habits of Legumes in Latvia

The survey was conducted to elucidate consumers’ eating preferences and expectations regarding legumes. Data from the survey was collected and summarized, and it is currently available online (https://www.visidati.lv/res/192061819327322/, accessed on 14 September 2023) in Latvia. In total, 776 self-recruited participants (82 males and 694 females) aged 17 to 80 years (mean = 46.3) participated in the survey. They consented to participate, confirming their understanding that their responses were confidential. Data were collected from 11 November 21, November 2022. The survey consisted of 15 questions, five of which were socio-demographic questions—gender, age, education level, marital status, and occupation—and ten of which were questions about the type, the consumption frequency, the attitude, and the choice motives of legumes used in the diet. The results of the survey were expressed as a percentage.

### 2.3. Materials

Crimson whole red lentils (*Lens culinaris* L.) were purchased wholesale; the country of origin is Canada. Nutritional value per 100 g of dry lentils: fat—1.1 g; carbohydrates—29.5 g; from them sugar—2.0 g; protein—25.8 g; and dietary fiber—30.5 g; the energy value of product per 100 g—1383 kJ/328 kcal.

Dry sauce Bolognese (vegan, with soya, gluten-free, and lactose-free) produced by GEFRO (Memmingen, Germany), purchased at an online grocery. Nutritional value per 100 g of sauce: fat—1.4 g; carbohydrates—8.8 g; from them sugar—6.0 g; protein—3.7 g; dietary fiber—2.2 g; and salt—1.6 g; the energy value of the product per 100 g—282 kJ/67 kcal.

Dry tomato soup (vegan, gluten-free, and lactose-free) produced by GEFRO (Memmingen, Germany), purchased at an online grocery. Nutritional value per 100 g of soup: fat—0.2 g; carbohydrates—3.3 g; from them sugar—1.7 g; protein—0.6 g; dietary fiber—0.6 g; and salt—0.41 g; the energy value of the product per 100 g—79 kJ/19 kcal.

### 2.4. Reagents

High-purity reagents and catalysts for liquid chromatography (HPLC-grade) were used for sample preparation. Concentrated hydrochloric acid (HCl, 37%) was purchased from VWR™ International (Paris, France); 25% ammonium hydroxide solution (NH_3_ × H_2_O); and potassium hydroxide (KOH, purity ≥ 85%) was acquired from Chempur (Piekary Śląskie, Silesia, Poland). Phenol (C_6_H_5_OH) (purity > 99%) and LC-MS-grade formic acid (HCOOH, puris r.a.) were purchased from Sigma-Aldrich (Darmstadt, Germany). A mixture of 17 amino acids (AAs) in liquid form with concentrations ranging from 1.25 to 2.6 µM mL was purchased from Merck KGaA (Darmstadt, Germany) and used as an external standard for constructing calibration curves. LC-MS-grade acetonitrile (MeCN) was ordered from the same source. Ultra-pure MS-grade isopropyl alcohol (C_3_H_8_O, purity ≥ 99.95%) was purchased from Biosolve Chimie (Dieuze, France). Ultra-pure water was produced using a reverse osmosis PureLab Flex Elga water purification system (Veolia Water Technologies, Paris, France).

### 2.5. Preparation of Lentils for Quick Meal Production

Lentils were soaked in water for 5 h at 12–15 °C. The soaked lentils were spread evenly on a pan and roasted for 15 min at a temperature of 160 ± 5 °C in a conveyer-type oven (Lincoln, NE, USA). The roasted lentils were cooled to room temperature. In the next step, the roasted lentils were chopped using Robot Coupe Blixer^®^4 (Clichy, France) with a speed of 3000 rpm and sifted through a sieve to obtain two fractions of chopped lentils—particle sizes greater than 1.6 mm and smaller than 1.6 mm. Division of lentils into fractions was essential in sensory evaluation to assess the effect of texture on liking. The roasted and chopped lentils were packed in the three-layer package and stored at room temperature until the start of further research.

### 2.6. Preparation of Roasted Lentils Quick Meals

Two types of quick meals were developed in this study: tomato soup with roasted lentils and roasted lentils with Bolognese sauce. The following quantities were used to prepare one portion of tomato soup: 30 g of roasted and chopped lentils and 12 g of dry tomato soup, and one portion of roasted lentils with Bolognese sauce: 42 g of roasted and chopped lentils and 28 g of dry Bolognese sauce. Approximately 200 mL of water (temperature 90–95 °C) was added to each blend with thoroughly mixing. Next, the containers with the prepared food were closed and endured for 5 min, before serving they were thoroughly mixed. 

### 2.7. Sensory Evaluation of Roasted Lentils Quick Meals

Three samples of tomato soup with roasted lentils and three samples of roasted lentils with Bolognese sauce were prepared with different combinations of lentil fractions using particle sizes greater than 1.6 mm and smaller than 1.6 mm (Table 1).

The degree of liking of sensory properties (color, flavor, and consistency) of 6 samples of roasted lentil quick meals (S1–S6 in Table 1) was determined using a 7-point hedonic scale (ISO 4121: 2003), where 1—extremely dislike; 4—neither like nor dislike; and 7—extremely like. Each panelist was served 20 g of warm (45–50 °C) coded quick meal samples. Warm black tea was used to neutralize the taste between the samples. Thirty panelists from Latvia University of Life Sciences and Technologies, students, and employees (aged 18 to 50) who use quick meals daily participated in the sensory evaluation. All participants gave their informed consent that they voluntarily participated in the sensory evaluation, that their responses were confidential, and that they could terminate their participation in the sensory evaluation without explanation. The scoring sheets, data collection, and data processing were carried out with FIZZ Acquisition 2.51 software (Biosystems, Couterno, France).

### 2.8. Analysis of Roasted Lentils and Roasted Lentils Quick Meals

The nutritional and energy values of roasted lentils and roasted lentil quick meals were analyzed in the accredited testing laboratory of J.S. Hamilton Poland Sp. according to the following methods and regulations: PB-116 ed. III of 11.08.2020 for protein (N × 6.25), AOAC 991.43: 1994 for dietary fiber, PB-287 ed. I of 27.09.2014 for total sugars after inversion, PB-286 ed. I of 26.09.2014 for fat, Regulation (EU) No. 1169/2011 of the European Parliament and of the Council of 25.10.2011 for energy value and carbohydrates, PN-A-79011-3:1998 for moisture, and PB-318/FAAS ed. I of 27.07.2015. for sodium, PN-EN 14122:2014-07 for thiamine; PN-EN 14152:2014-07 for riboflavin; PB-327 ed. 2 of 05.09.2022 [EN] for folate; PB-223/ICP ed. II of 12.01.2015 for calcium; PB-223/ICP ed. II of 12.01.2015 for potassium; PB-223/ICP ed. II of 12.01.2015 for magnesium; PB-223/ICP ed. II of 12.01.2015 for phosphorus; PB-223/ICP ed. II of 12.01.2015 for iron.

### 2.9. The Release of Amino Acids Using Acid-Assisted Hydrolysis

The release of AAs from the prepared product matrix was performed according to the ISO 13903:2005 protocol with modifications, subjecting the sample to acid hydrolysis with 5 mL of 6M HCl solution. Hydrolysis was conducted in 22.0 mL glass Headspace chromatography vials (PerkinElmer, Inc., Waltham, MA, USA) with screw caps and silicone seals. The prepared sample (200 mg ± 0.1) was aged for 24 h in a drying cabinet “Pol-Eko Aparatura SP.J.,” (Wodzislava Slonska, Poland) at a temperature of 110 °C. During hydrolysis, the stabilizing reagent phenol, added directly to the sample at 0.02% (*w/w*), was used to delay the oxidation-reduction reaction of the compounds of interest. The volume of the hydrolysate obtained after hydrolysis was brought up to 10.0 mL with H_2_O, and the pH of the medium was normalized to 6.5–6.8 using 2.50 mL of 25% ammonium hydroxide solution. Finally, the volume of the obtained hydrolysate was brought to 14.0 mL and subjected to intensive mixing for 1 min with a “ZX3” vortex mixer (Velp^®^ Scientifica, Usmate Velate, Italy). For the separation of fractions, the prepared hydrolysates were centrifuged for 10 min at 16,070× *g* and 19.0 ± 1 °C using a “Hermle Z 36 HK” centrifuge (Hermle Labortechnik, GmbH, Wehingen, Germany), and the upper organic layer was collected. Before LC-MS analysis, the collected layer was filtered through a 0.22 µm polytetrafluoroethylene (CROMAFIL^®^ Xtra H-PTFE) hydrophilized membrane filter (Macherey-Nagel GmbH & Co. KG, Dueren, Germany).

### 2.10. HPLC-ESI-TQ-MS/MS Analytical Conditions for Amino Acid Determination

AA analysis was accomplished using a “Shimadzu Nexera UC” series liquid chromatograph (LC) (Shimadzu Corporation, Tokyo, Japan) in tandem with a triple quadrupole mass selective detector (TQ-MS-8050, Shimadzu Corporation, Tokyo, Japan) equipped with electrospray ionization (ESI). Chromatographic separation of AAs was performed using a reversed-phase “Discovery^®^ HS F5-3” column (3.0 µm, 150 × 2.1 mm, Merck KGaA, Darmstadt, Germany) at a temperature of 40 °C and a flow rate of mobile phase 0.25 mL min^−1^. The sample injection of 3.0 μL was conducted automatically, and the loop was rinsed with 2-propanol. Mobile phase composition: acidified H_2_O (0.1% HCOOH *v*/*v*) (A) and acidified MeCN (0.1% HCOOH *v*/*v*) (B). The stepwise gradient elution program of mobile phase B during 15 min was programmed as follows: T_0_ min = 5.0%, T_2_ min = 5.0%, T_7.0_ min = 30.0%, T_11.0_ min = 60.0%, T_12.0_ min = 80.0%, and T_12.1_ min = 5.0%. After each analysis, re-equilibration was completed for 3 min following the initial gradient conditions. A MeCN injection was included after each sample as a blank to avoid the carry-over effect. Data were acquired using “LabSolutions Insight LC-MS” version 3.7 SP3, which was also used for LC-MS control and data processing. This study used the positive electroionization mode, while data were collected in profile and centroid modes with a data storage threshold of 5000 absorption MS. Operating conditions of the mass-selective detector: detector voltage 1.98 kV, conversion dynode voltage 10.0 kV, interface voltage 4.0 kV, interface temperature 300 °C, desolvation line temperature 250 °C, heating block temperature 400 °C, spray gas—argon (Ar, purity 99.9%) with a flow rate of 3.0 L min^−1^; heating gas—carbon dioxide (CO^2^, purity 99.0%) with a flow rate of 10.0 L min^−1^, and drying gas—nitrogen (N^2^, separated from air using a nitrogen generator “Peak Scientific Instruments Ltd.” (Inchinan, Scotland, UK), purity 99.0%) with a flow rate of 10.0 L min^−1^. AAs were determined using the selected ion transitions (multiple reaction monitoring—MRM) after being acquired and optimized by the LC-MS system. Quantitative AAs analysis was executed at 15 °C by injecting 3.0 μL of a calibration solution in the concentration range from 0.075 to 2.5 μM L^−1^. The stock standard solution was prepared just before the analysis. Selective ion chromatogram (TIC) in MRM mode represents 17 AAs and one tryptophan derivative in Figure 2.

## 3. Results

### 3.1. Survey on Consumption Habits of Legumes in Latvia

Canned beans are the most popular product among Latvian residents, where taste and quality are the determining factors when choosing legumes (Table 2).

The respondents mainly associated legumes with a healthy diet and a feeling of satiety for a longer period of time. Moreover, more than half of respondents (55.9%) confirmed that they would like to increase the consumption of legumes in the next 2–3 years. This study’s findings align with the study conducted in Sweden [8], which could mean that the demand for legume products/meals could increase if products/meals are offered according to the needs of consumers.

### 3.2. Sensory Evaluation of Roasted Lentils Quick Meals

Not only the overall appearance (color, flavor) of the product but also the consistency are important for consumers. Three samples of tomato soup with roasted lentils and three roasted lentils with Bolognese sauce were prepared with different grinds of roasted lentils and the added amount, which were evaluated for their sensory properties (Figure 3).

The panelists preferred the sensory properties—color; flavor; and consistency—of roasted lentils with Bolognese sauce; and there were no significant differences (*p* ≥ 0.05) in the liking between them. The samples of tomato soup with roasted lentils were recognized as less pleasant compared to the samples of roasted lentils with Bolognese sauce. There were no significant differences (*p* ≥ 0.05) in liking the flavor and consistency of the samples of tomato soup with roasted lentils. The panelists said they liked the color of the roasted lentils with Bolognese sauce, despite the fact that there were large pieces of lentils in the texture that looked undercooked. The panelists noted that the large, unswollen pieces of lentils in the tomato soup with roasted lentils were difficult to chew and gave the soup an uneven consistency. The sample S1, which contained 72% chopped lentils with particle size < 1.6 mm, was recognized as the least pleasant in terms of color, although it allowed to obtain a tomato soup with a more pronounced color than in the case of using finely chopped (>1.6 mm) lentils, which swelled more easily. The best-evaluated samples in both types of quick meals with roasted lentils, namely S3 and S5, were selected for further investigation by chemical and nutritional analysis.

### 3.3. Nutritional and Energy Value of Roasted Lentils and Roasted Lentils Quick Meals

The uniqueness of lentils lies in their nutritional value, being high in protein and low in fat [14], which was also preserved in roasted lentils quick meals (Table 3).

Roasted lentil quick meals have a low energy value due to their low fat and relatively low carbohydrate content, but they are high in protein, providing from 15.6% to 26.2% of the daily protein requirement per serving.

### 3.4. The Composition of Amino Acids, Vitamins, and Minerals in Roasted Lentils and Roasted Lentils Quick Meals

Legumes are a relatively underutilized source of high-quality dietary proteins worldwide, while the shift from animal proteins to legumes would allow for a reduction in greenhouse gas emissions [21]. Legumes represent the full spectrum of amino acids (AAs), especially the availability of essential (EAAs) and branched-chain (BCAAs) representatives, making them promising alternatives to those individuals with increased physical activity [22]. The dominance of Arg, Asp, Glu, and Leu in lentils has been highlighted by Khazaei et al. [23], and their amount contributes to more than half of total AAs. A similar observation has been made in the current research, where the content of the AAs mentioned above corresponds to 47.3% of the total AAs (Table 4).

The availability of Glu, followed by Asp, Leu, Arg, Lys, Ser, and Val, was revealed in roasted lentils and the observed values were to a greater extent consistent with those reported for red baked lentils by Nosworthy et al. [24]. It is worth noting that the content of branched-chain AAs in roasted lentils was found to be 3.83 g/100 g^−1^, which is in complete agreement with the data reported by Nosworthy et al. [24]. According to data obtained, lentils are limited in such AAs as Met and Cys, which Mookerjee & Tanaka [25] observation reinforces. One-third of the AAs observed in roasted lentils are EAAs, corresponding to 8.16 g 100 g^−1^ DW. Thus, an adequate intake of roasted lentils ensures sufficient amounts of both EAAs and BCAAs, which are crucial to individuals’ adhesion to veganism and plant-based diets. A similar descending order of AAs was found in developed roasted lentils quick meals, where the dominance of Glu, followed by Asp, Leu, Arg, Lys, Ser, and Val, can be seen.

The content of B-group vitamins depends on the location of the lentils as the cotyledons are rich in B_1_ and B_3_ vitamins, while the lentils coats are rich in B_2_, B_5_, B_6_, and B_9_ vitamins [26]. The content of vitamins B_1_, B_2_, and B_9_ was evaluated in roasted lentils and their quick meals (Table 5). Quantitatively, lentils contain the most vitamin B_1_ (0.51–0.67 mg 100 g^−1^) [27], the content of which significantly decreased after roasting of lentils, which is confirmed by the results of the Lui et al. [27] study that thermal treatment reduces the level of B group vitamins in lentils.

The most abundant minerals in lentils are potassium, phosphorus, magnesium, and calcium [28] which were also preserved in roasted lentils and their quick meals. The range of minerals in roasted lentils and quick meals studied was as follows: potassium, phosphorus, magnesium, calcium, and iron. The level of vitamins and minerals in ready-roasted lentil quick meals depends on the amount of added roasted lentils; therefore, the ready-quick meals of roasted lentils with Bolognese sauce showed a higher level of vitamins and minerals.

## 4. Discussion

The consumption of legumes in Latvia is low, according to the research data from Sweden survey [8] and in Europe as a whole [3], even though a large number of respondents associated the use of legumes with a healthy choice, stating that it allows them to eat or snack healthier. Surprisingly, most respondents (83.4%) disagreed that legumes could replace or exclude meat from the diet. It shows how deep–rooted the meat-eating traditions are in the Latvian population, which is confirmed by statistical data: 36.5% of respondents consumed meat 3–5 days a week, 37.1%—consumed meat 1–2 days a week; 10.4%—6–7 days a week [29]. It is challenging to convince the public to reduce meat consumption and include legumes as an alternative. The favourite choice of Latvian respondents was beans (72.4%), while lentils were the least popular product, as 1/3 of respondents stated that they do not eat them in general. However, as a promising fact, it should be mentioned that 55.9% of respondents expressed their readiness to increase the consumption of legumes in the next 2–3 years, which indicates a potential demand for quality legume products.

Food market analysis revealed a significant increase in the consumption of healthy snacks, which is expected to grow by 6.6% annually, reaching 152.3 billion US dollars in 2030 [30]. On the one hand, it can be evaluated positively because people are paying more and more attention to a healthy diet; on the other hand, it should be noted that snacks, although healthy, cannot make up a person’s meal. This means that there is a need for healthy, delicious, and quick meals because people do not want to spend time preparing food every day. This, in turn, opens up the opportunity for the roasted lentils quick meals developed in this study to provide warm, quick meals with an essential protein provision of 15.6% (tomato soup with roasted lentils) and 26.2% (roasted lentils with Bolognese sauce) of daily reference intake for protein. In prepared quick meals, i.e., tomato soup with roasted lentils, the content of individual AAs, particularly Glu, was substantially higher than that observed in the raw material. The increase in Glu is conditioned by additional AA-rich dry ingredients incorporated into the recipe, such as tomatoes, in which Glu predominates and is reported to have an umami taste [31,32]. The content of BCAAs in prepared tomato soup with roasted lentils corresponded to 2.91 g 100 g^−1^ DW, while in roasted lentils with Bolognese sauce, it corresponded to 3.05 g 100 g^−1^ DW. Like roasted lentils, one-third of the AAs in prepared lentils-based quick meals are EAAs, with roasted lentils with tomato soup having the lowest amount and roasted lentils with Bolognese sauce the highest. The total content of AAs was considerably lower than found in raw material, corresponding to 18.60 and 18.69 g 100 g^−1^ DW in tomato soup with roasted lentils and roasted lentils with Bolognese sauce, respectively. The decrease in the content of AAs is due to added ingredients with a lower AA content than lentils. Overall, observed AA values in prepared quick meals make them promising alternatives to meat and fish products while contributing more positively to environmental sustainability.

Along with the beneficial amino acids’ composition, the quick meals of roasted lentils showed a favourable level of vitamins and minerals. One portion of ready-roasted lentils with Bolognese sauce provided 15.5% of the daily reference intake of thiamin and 17.6% of the daily reference intake of vitamin B9 [20], which can be considered a significant amount. The riboflavin content in ready-quick meals was above 15% of daily reference intake. According to the daily reference intake, one portion of a ready quick meal of tomato soup with roasted lentils covered 27.0% potassium, 19.9% phosphorus, 11.3% magnesium, 3.7% calcium, and 20.3% iron. In comparison, one portion of ready quick-meal of roasted lentils with Bolognese sauce provided 46.6% of potassium, 31.0% of phosphorus, 20.5% of magnesium, 7.9% of calcium, and 25.6% of iron according to daily reference intake [20]. Iron content in roasted lentils quick meals can be seen as promising, considering the issue of iron deficiency among the population. WHO reported that over 38 K non-pregnant women aged 15–49 suffer from iron deficiency [33]. However, there is needed further studies about the bioavailability of quick meals because the anti-nutritional properties of lentils, such as trypsin inhibitory, as well as anti-nutrients, such as phytic acid and tannins, need to be addressed while legume-based product development stage [16,34]. Mayer Labba’s study concluded that fava bean meal had higher iron content than beef and cod protein meals; however, iron absorption from the fava bean meal was low due to the high level of phytic acids [35]. Anti–nutrients activity could be limited by roasting lentils [12,16]. However, it would be necessary to evaluate the antinutritional properties after roasting because the time and temperature of heat treatment can affect the result in different ways.

## 5. Conclusions

Lentils are considered underutilized or neglected crops as they receive little attention in the food industry and agriculture. However, the need to improve cropping systems while considering the steadily rising demand for plant-based protein-rich products has led researchers worldwide to develop new legume-based products. The results revealed that consumers associate legumes with satiety. Most respondents highlighted that due to health-promoting benefits, they would increase the consumption of legumes in the next few years. According to the sensory evaluation results, panellists preferred the developed roasted lentils with Bolognese sauce as the colour, flavour, and consistency were the most acceptable compared with the other products. The two legume-based quick meal products, i.e., tomato soup with roasted lentils and roasted lentils with Bolognese sauce, demonstrated low fat and high protein content, providing 15.6% to 26.2% of the daily protein requirement per serving. The amount of vitamins and minerals such as thiamin, folate, potassium, phosphorus, magnesium and calcium was substantially higher in roasted lentils with Bolognese sauce. The chromatographic fingerprinting revealed the relative abundancy of amino acids in the developed products, including essential and branched-chain amino acids. Overall, the developed lentil-based quick meal products could be treated as a sustainable product, which in the long run could affect consumers’ attitude towards lentils as a meat protein alternative. Given the favorable results, further studies on the bioavailability of compounds in quick meals must be carried out to claim that they can be equally as nutritious as meat products.

## Figures and Tables

**Figure 1 foods-13-00099-f001:**
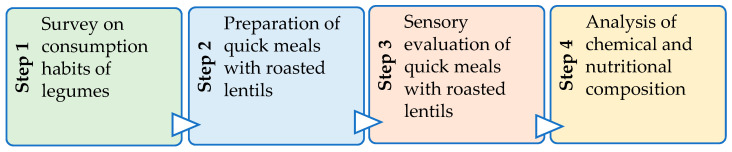
Schematic representation of the study design.

**Figure 2 foods-13-00099-f002:**
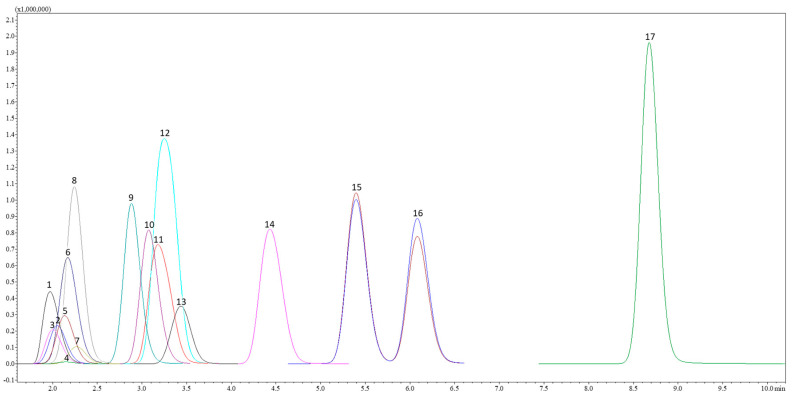
Selected ion chromatogram in MRM mode represents the profiles of 17 amino acids in a standard solution at a concentration of 2.5 μM L^−1^. Note: 1—Cystine (Cys); 2—Serine (Ser); 3—Aspartic acid (Asp); 4—Glycine (Gly); 5—Threonine (Thr); 6—Glutamic acid (Glu); 7—Alanine (Ala); 8—Proline (Pro); 9—Histidine (His); 10—Lysine (Lys); 11—Valine (Val); 12—Arginine (Arg); 13—Methionine (Met); 14—Tyrosine (Tyr); 15—Isoleucine (Ile); 16—Leucine (Leu); 17—Phenylalanine (Phe).

**Figure 3 foods-13-00099-f003:**
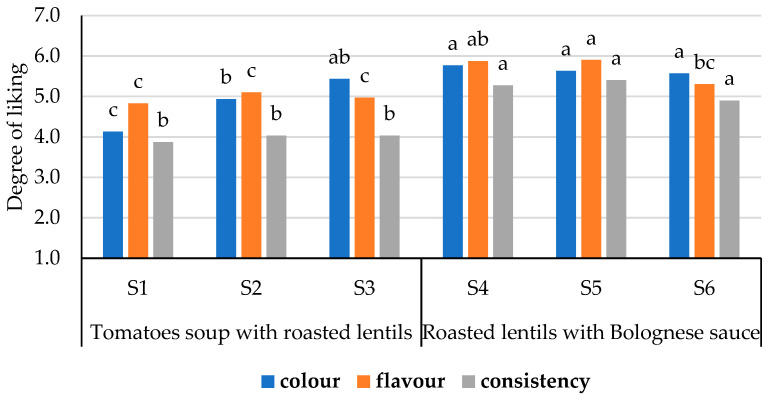
Degree of liking of quick meals sensory properties. Different letters indicate significant differences between samples sensory properties (*p* ≤ 0.05).

**Table 1 foods-13-00099-t001:** Ingredient ratio for roasted lentil quick meals, %.

Sample Code	Dry Tomato Soup	Dry Bolognese Sauce	Roasted Chopped Lentils	
			<1.6 mm	>1.6 mm
S1	28	-	72	-
S2	28	-	36	36
S3	28	-	-	72
S4	-	40	60	-
S5	-	40	20	40
S6	-	40	40	20

**Table 2 foods-13-00099-t002:** The results of the survey on the consumption habits of legumes in Latvia, %.

Questions		I Eat Regularly		I Rarely Eat	I Don’t Eat
Which legumes do you eat?	Peas	62.9		36.0	1.1
	Beans	72.4		27.1	0.5
	Lentils	24.7		42.4	32.9
		**I choose regularly**		**I rarely choose**	**I don’t choose**
What legumes do you choose for your diet?	Fresh	36.4		51.5	12.1
	Canned	65.7		29.1	5.2
	Boiled	31.9		30.7	37.5
	Roasted	6.5		30.5	62.9
	Semi-finished	17.7		31.9	50.5
	Instant	21.9		33.5	44.6
		**Important**		**Semi important**	**Unimportant**
Evaluate the importance of the mentioned factors when purchasing legumes for your diet!	Price	56.6		31.1	12.4
	Made in Latvia	58.6		30.5	10.9
	Allergic	30.0		8.0	62.0
	Quality	93.0		5.4	1.6
	Taste	96.2		2.8	1.1
	Energy value	47.1		28.6	24.3
	No GMO	72.5		16.1	11.4
	Organic	55.6		28.2	16.2
	Environmental impact	50.1		30.8	19.1
	Brand	57.3		30.4	12.4
		**YES**		**NO**	**Maybe**
Would you like to increase the consumption of legumes in your diet in the next 2–3 years?		55.9		9.1	34.5
		**I agree**	**I rather agree**	**I rather disagree**	**I disagree**
The inclusion of legumes into my diet gives me an opportunity…	To exclude animal products from the diet	4.5	13.3	31.9	50.3
	To replace part of animal products in the diet	17.5	28.4	21.0	33.1
	To cook a meal faster	16.2	31.6	31.1	21.1
	To eat healthier	47.4	44.0	5.4	3.2
	To feel full for a longer time	36.3	45.3	12.9	5.6
	To snack healthier	29.9	38.9	16.9	14.3
	To reduce the overweight	13.9	31.2	32.9	21.9
	To increase the muscle mass	11.2	33.7	31.7	23.4
	To exclude some meal	5.7	16.3	35.7	42.2
	To be environmentally friendly	22.4	40.4	21.2	15.9
	To be contemporary	8.4	15.9	27.5	48.2

**Table 3 foods-13-00099-t003:** Nutritional and energy value of roasted lentils and roasted lentils quick meals.

	Roasted Lentils	Tomato Soup with Roasted Lentils per 100 g DW)	Reference Intake ^1^ per 240 g of Ready Meal, %	Roasted Lentils with Bolognese Sauce per 100 g DW	Reference Intake ^1^ per 270 g of Ready Meal, %
Moisture, g	6.5 ± 0.2	5.6	-	4.1	-
Fat, g	1.7 ± 0.3	1.4 ± 0.2	0.9	1.5 ± 0.2	1.6
Saturated fatty acids, g	0.1 ± 0.0	0.3 ± 0.0	0.5	0.3 ± 0.1	1.0
Carbohydrates, g	41.1 ± 1.1	31.5 ± 0.7	5.1	28.7 ± 0.8	7.7
Sugars, g	4.8 ± 0.4	4.2 ± 0.2	2.0	5.9 ± 0.2	4.6
Dietary Fiber, g	22.4 ± 0.9	5.8 ± 0.3	-	5.9 ± 0.2	-
Protein, g	26.0 ± 1.1	18.6 ± 0.95	15.6	18.7 ± 0.8	26.2
Salt, g	0.09 ± 0.01	0.2 ± 0.0	1.3	0.7 ± 0.1	8.3
Energy value, kJ	1383 ± 41	950 ± 28	4.8	909 ± 30	7.6

Note: Values are means ± SD (*n* = 5). DW—dry weight. ^1^ Regulation (EU) No 1169/2011 of the European Parliament and of the Council of 25 October 2011 on the provision of food information to consumers, Annex XIII Part B—reference intakes for energy and selected nutrients (adults) [20]. The amounts of 270 g and 240 g correspond to one portion.

**Table 4 foods-13-00099-t004:** The composition of amino acids in roasted lentils and roasted lentils quick meals, g 100 g^−1^ DW.

Amino Acids	Roasted Lentils		Tomato Soup with Roasted Lentils		Roasted Lentils with Bolognese Sauce	
	g 100 g^−1^ DW	% of Total Amino Acids	g 100 g^−1^ DW	% of Total Amino Acids	g 100 g^−1^ DW	% of Total Amino Acids
Cystine (Cys)	0.29 ± 0.02	1.26	0.27 ± 0.01	1.44	0.28 ± 0.01	1.47
Serine (Ser)	1.42 ± 0.11	6.26	1.05 ± 0.08	5.63	1.08 ± 0.05	5.80
Aspartic acid (Asp)	3.13 ± 0.28	13.75	2.29 ± 0.04	12.31	2.35 ± 0.07	12.58
Glycine (Gly)	1.02 ± 0.05	4.48	0.78 ± 0.01	4.21	0.85 ± 0.00	4.54
Threonine (Thr)	1.08 ± 0.08	4.76	0.84 ± 0.09	4.51	0.84 ± 0.04	4.51
Glutamic acid (Glu)	4.32 ± 0.29	18.95	4.46 ± 0.14	24.00	4.03 ± 0.03	21.57
Histidine (His)	0.51 ± 0.01	2.25	0.36 ± 0.03	1.92	0.35 ± 0.06	1.89
Alanine (Ala)	1.10 ± 0.00	4.82	0.92 ± 0.08	4.94	0.95 ± 0.07	5.06
Proline (Pro)	1.11 ± 0.04	4.89	0.87 ± 0.05	4.66	0.91 ± 0.03	4.84
Arginine (Arg)	1.66 ± 0.07	7.28	1.26 ± 0.09	6.77	1.30 ± 0.05	6.97
Lysine (Lys)	1.52 ± 0.02	6.69	1.16 ± 0.10	6.22	1.19 ± 0.04	6.38
Valine (Val)	1.18 ± 0.02	5.19	0.90 ± 0.08	4.84	0.91 ± 0.10	4.89
Methionine (Met)	0.14 ± 0.01	0.63	0.16 ± 0.01	0.85	0.11 ± 0.01	0.60
Tyrosine (Tyr)	0.56 ± 0.03	2.48	0.43 ± 0.06	2.32	0.40 ± 0.04	2.11
Leucine (Leu)	1.68 ± 0.01	7.36	1.27 ± 0.04	6.82	1.37 ± 0.07	7.33
Isoleucine (Ile)	0.97 ± 0.00	4.25	0.74 ± 0.03	3.98	0.77 ± 0.08	4.11
Phenylalanine (Phe)	1.07 ± 0.05	4.71	0.85 ± 0.01	4.57	0.10 ± 0.03	5.36
BCAAs	3.83 ± 0.03	16.81	2.91 ± 0.15	15.64	3.05 ± 0.25	16.33
EAAs	8.16 ± 0.19	35.84	6.27 ± 0.39	33.71	6.55 ± 0.43	35.06
Total	22.77 ± 1.09	100.00	18.60 ± 0.95	100.00	18.69 ± 0.77	100.00

Note: Values are means ± SD (*n* = 5). DW—dry weight; BCAAs—branched chain amino acids (leucine, isoleucine, valine); EAAs—essential amino acids (threonine, histidine, lysine, valine, methionine, leucine, isoleucine, phenylalanine).

**Table 5 foods-13-00099-t005:** The composition of vitamins and minerals in roasted lentils and roasted lentils quick meals.

Vitamin/Mineral	Roasted Lentils, mg 100 g^−1^ DW	Tomato Soup with Roasted Lentils, mg 100 g^−1^ DW	Tomato Soup with Roasted Lentils, mg 240 g^−1^ of Ready Meal	Roasted Lentils with Bolognese Sauce, mg 100 g^−1^ DW	Roasted Lentils with Bolognese Sauce, g 270 g^−1^ of Ready Meal
Thiamin	0.28 ± 0.01	0.28 ± 0.01	0.12 ± 0.01	0.24 ± 0.01	0.17 ± 0.00
Riboflavin	0.20 ± 0.01	0.22 ± 0.01	0.09 ± 0.01	0.21 ± 0.00	0.15 ± 0.00
Folate	35.30 ± 1.23 ^1^	47.40 ± 2.01 ^1^	19.91 ± 0.87 ^2^	50.40 ± 2.56 ^1^	35.28 ± 1.29 ^3^
Calcium	51.40 ± 2.12	69.80 ± 3.24	29.32 ± 1.02	90.20 ± 4.67	63.14 ± 2.89
Potassium	781.00 ± 25.01	1288.00 ± 76.00	540.96 ± 18.70	1330.00 ± 79.44	931.00 ± 27.68
Magnesium	94.40 ± 9.00	101.00 ± 16.71	42.42 ± 3.86	110.00 ± 15.79	77.00 ± 9.76
Phosphorus	346.00 ± 26.00	331.00 ± 18.00	139.02 ± 6.98	310.00 ± 17.64	217.00 ± 14.86
Iron	7.15 ± 0.98	6.77 ± 0.65	2.84 ± 0.09	5.11 ± 0.47	3.58 ± 0.10

Note: Values are means ± SD (*n* = 5). DW—dry weight. ^1^ µg 100 g^−1^; ^2^ µg 240 g^−1^; ^3^ µg 270 g^−1.^

## Data Availability

The data presented in this study are available on request from the corresponding author. Data is contained within the article (and Appendix A).

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
