# Peer review of "The Evaluation of Roasted Lentils (L. culinaris L.) Quick Meals as An Alternative to Meat Dishes"

_foods, 2023, doi:10.3390/foods13010099_

Round 1
Reviewer 1 Report
Comments and Suggestions for Authors
Introduction
Line 33: “The consumption of legumes in Europe is low compared to other regions.” Provide reference.
Despite the nutrient content, provide justification about why promoting lentil in East Europe where lentil is not commonly consumed.
What are the research questions for the survey, recipe design and nutrient analysis? What are the study designs? Why did the study choose the two recipes with tomato soup and Bolognese sauce?
What is the definition of quick meals?
It seems the survey work is an important part of the study, but the manuscript did not provide background for the survey.
Methods:
Methods and objectives of the survey were not mentioned in the Method section at all.
What is the purpose of measuring nutrient contents of the two recipes? There is no relation between the survey work and nutrient analysis.
Conclusion:
I do not think simply comparing two type of recipes (with tomato soup and Bolognese sauce) can make such conclusion:
“The developed roasted lentils quick-meals can be evaluated as a sustainable product and promising alternative to increase the consumption of lentils, meet vegan diet requirements and replace meat dishes due to the composition of amino acids.”
Comments on the Quality of English LanguageEnglish is fine.
Author Response
Dear Reviewer,
Thanks for all your comments.
Introduction:
- Line 33: “The consumption of legumes in Europe is low compared to other regions.” Provide reference.
Authors’ response: The reference was added in the manuscript.
- Despite the nutrient content, provide justification about why promoting lentil in East Europe where lentil is not commonly consumed.
Authors’ response: The aim of the study was to develop alternative to animal proteins nutrient-dense plant-based quick meals using roasted lentils as basic raw material, performing sensory analysis and evaluating the content of amino acids, minerals, and vitamins. The study did not set out to explain why the consumption of lentils in Eastern Europe is low, as this would require a multifaceted study to explain it.
- What are the research questions for the survey, recipe design and nutrient analysis? What are the study designs? Why did the study choose the two recipes with tomato soup and Bolognese sauce?
Authors' response: The authors have added the study design in the methodology section, which explains the study's progress and answers unclear questions about why such quick meals have been chosen.
- What is the definition of quick meals?
Authors’ response: A quick meal is a meal which is easy to prepare and/or cooked in short time. In the case of this study, the preparation of quick meals with roasted lentils required hot water, which was added to blend with thorough mixing. Next, the containers with the prepared food were closed and endured for 5 min, before serving they were thoroughly mixed.
- It seems the survey work is an important part of the study, but the manuscript did not provide background for the survey.
Authors’ response: The added the study design in the methodology section provides the background for the survey. First, it was important to understand the consumption habits of legumes, then quick meals with roasted lentils were developed, followed by sensory evaluation. Chemical and nutritional analyses were performed on the quick meal samples that panellists have highlighted as the most preferable.
Methods:
- Methods and objectives of the survey were not mentioned in the Method section at all.
Authors’ response: The objective of the survey is indicated on Lines 83-84 followed by the survey methodology - description. Specific questions can be found in the results section in Table 1.
- What is the purpose of measuring nutrient contents of the two recipes? There is no relation between the survey work and nutrient analysis.
Authors’ response: The authors have added the study design in the methodology section, which explains the progress of the study, provides answers to unclear questions, why such quick meals have been chosen and further analyzed for chemical and nutritional composition.
Conclusion:
- I do not think simply comparing two types of recipes (with tomato soup and Bolognese sauce) can make such conclusion: “The developed roasted lentils quick-meals can be evaluated as a sustainable product and promising alternative to increase the consumption of lentils, meet vegan diet requirements and replace meat dishes due to the composition of amino acids.”
Authors’ response: The authors made corrections to the conclusions.
Reviewer 2 Report
Comments and Suggestions for Authors
The paper is interesting and I find suitable. However it is necessary to include the statistic in the tables and figures, to have same style and colour in figures and give more details to results especially in the correlation between composition and sensory aspect.
I consider the paper should be revised after major revision.
Comments on the Quality of English LanguageThe paper is interesting and I find suitable for publications. However it is necessary to incluide the statistic in the tables and figures, to have same style and colour in figures and give more details to results especially in the corelation between composition and sensory aspect.
I consider the paper can be publish after major revision
Author Response
Thanks for your comment.
- The paper is interesting and I find suitable. However, it is necessary to include the statistic in the tables and figures, to have same style and colour in figures and give more details to results especially in the correlation between composition and sensory aspect.
Authors’ response: Statistics are included where permissible, i.e., in sensory evaluation, where three samples of tomato soup with roasted lentils and three samples of roasted lentils with Bolognese sauce were prepared with different combinations of lentils fractions using particle size greater than 1.6 mm and smaller than 1.6 mm. After sensory evaluation, two different quick meals with the best rating were selected for chemical and nutritional analyses. These samples are not statistically comparable because one is a soup, and the other is a second dish with different lentil content and portion output. Figure 2 reflects the selected ion chromatogram in MRM mode extracted from the software and unfortunately cannot be changed. The rest of the Figures are designed according to the requirements. The authors have added the study design in the methodology section, which explains the study's progress and answers unclear questions about the sensory study and its effect on its progress.
Round 2
Reviewer 1 Report
Comments and Suggestions for Authors
The study tackles the crucial topic of healthy and sustainable consumption. The authors have effectively addressed the reviewers' concerns. The manuscript merits publication.
Reviewer 2 Report
Comments and Suggestions for Authors
I consider the paper can be accepted